# Learning Geometric Representations of Interactive Objects

## Abstract

We address the problem of learning geometric representations from observations perceived by an agent operating within an environment and interacting with an external object. To this end, we propose a representation learning framework that extracts the state of both the agent and the object from unstructured observations of arbitrary nature (e.g., images). Supervision comes from the performed actions alone, while the dynamics of the object is assumed to be unknown. We provide a theoretical foundation and formally prove that an ideal learner is guaranteed to infer an isometric representation, disentangling the agent from the object. Finally, we investigate empirically our framework on a variety of scenarios. Results show that our model reliably infers the correct representation and outperforms vision-based approaches such as a state-of-the-art keypoint extractor.

## 1 Introduction

A fundamental aspect of intelligent behavior by part of an agent is building rich and structured *representations* of the surrounding world (Ha & Schmidhuber (2018)). Through structure, in fact, a representation can potentially achieve semantic understanding, efficient reasoning and generalization (Lake et al. (2017)). However, in a realistic scenario an agent perceives unstructured and high-dimensional observations of the world (e.g., images). The ultimate goal of inferring a representation consists thus of extracting structure from such observed data (Bengio et al. (2013)). This is challenging and in some instances requires supervision or biases. For example, it is known that *disentangling* factors in data is mathematically impossible in a completely unsupervised way (Locatello et al. (2019)). In order to extract structure, it is thus necessary to design methods and paradigms relying on additional information and specific assumptions.

In the context of an agent interacting with the world, a fruitful source of information is provided by the actions performed and collected together with the observations. Based on that, several recent works have explored the role of actions in representation learning and proposed methods to extract structure from interaction (Kipf et al. (2019); Mondal et al. (2020); Park et al. (2022)). The common principle underlying this line of research is encouraging the representation to replicate the effect of actions in a structured space – a property referred to as *equivariance* [1]. In particular, it has been shown in Marchetti et al. (2022) that equivariance enables to extract the internal state of the agent (i.e., its pose in space), resulting in a lossless and geometric representation. The question of how to represent features of the world which are extrinsic to the agent (e.g., objects) has been left open. Such features are dynamic since they change as a consequence of interaction. They are thus challenging to capture in the representation but are essential for understanding and reasoning by part of the agent.

In this work we consider the problem of learning a representation of an agent together with an external object the agent interacts with (see Figure 1). We focus on a scenario where the object displaces only when it comes in contact with the agent, which is realistic and practical. This enables to design a representation learner that attracts the representation of the object to the one of the agent when interaction happens or keeps it invariant otherwise. Crucially, we make no assumption on the complexity of the interaction: the object is allowed to displace arbitrarily and its dynamics is unknown. All the

---

[1] Alternative terminologies from the literature are *World Model* Kipf et al. (2019) and *Markov Decision Process Homomorphism* (van der Pol et al. (2020)).

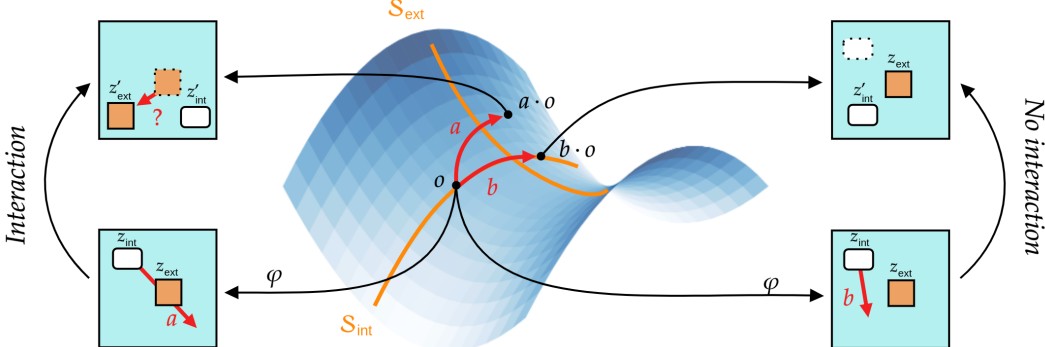

Figure 1: Our framework enables to learn a representation $\varphi$ recovering the geometric and disentangled state of both an agent ($z_{\text{int}}$, white) and an interactable object ($z_{\text{ext}}$, brown) from unstructured observations $o$ (e.g., images). The only form of supervision comes from actions $a, b$ performed by the agent, while the transition of the object (question mark) in case of interaction is unknown. In case of no interaction, the object stays invariant.

losses optimized by our learner rely on supervision from interaction alone i.e., on observations and performed actions. This makes the framework general and in principle applicable to observations of arbitrary nature. We moreover provide a formalization of the problem and theoretical grounding for the method. Our core theoretical result guarantees that the representation inferred by an ideal learner recovers both the ground-truth states up to a translation. This implies that the representation is isometric (i.e., fully extracts the geometric structure of states) and disentangles the agent from the object. As a consequence, the representation preserves the geometry of the state space underlying observations. The preservation of geometry makes the representation lossless, interpretable and disentangled. We empirically show that our representations not only outperform in quality of structure a state-of-the-art keypoint extractor, but can be leveraged by a downstream learner in order to solve control tasks efficiently. In summary our contributions include:

- A representation learning framework extracting geometry from observations in the context of an agent interacting with an external object.
- A theoretical result guaranteeing that the above learning framework, when implemented by an ideal learner, infers an isometric representation for data of arbitrary nature.
- An empirical investigation of the framework on a variety of environments with comparisons to computer vision approaches (i.e., keypoint extraction) and applications to a control task.

We provide Python code implementing our framework together with all the experiments as part of the supplementary material.

## 2    RELATED WORK

**Equivariant Representation Learning.**    Several recent works have explored the idea of incorporating interactions into representation learning. The common principle is to infer a representation which is equivariant i.e., such that transitions in observations are replicated as transitions in the latent space. One option is to learn the latent transition end-to-end together with the representation (Kipf et al. (2019); van der Pol et al. (2020); Watter et al. (2015)). This approach is however non-interpretable and the so-obtained representations are not guaranteed to extract any structure. Alternatively, the latent transition can be designed a priori. Linear and affine latent transitions have been considered in Guo et al. (2019), Mondal et al. (2020) and Park et al. (2022) while transitions defined by (the multiplication of) a Lie group have been discussed in Marchetti et al. (2022), Mondal et al. (2022). As shown in Marchetti et al. (2022), for static scenarios (i.e., with no interactive external objects) the so-obtained representations are structured and completely recover the geometry of the underlying state of the agent. Our framework adheres to this line of research by modelling the latent transitions via the additive Lie group $\mathbb{R}^n$. We however further extend the representation to

include external objects. Our framework thus applies to more general scenarios and dynamics while still benefiting from the geometrical guarantees.

**Keypoint Extraction.** When observations are images computer vision offers a spectrum of classical approaches to extract geometric structure. In particular, extracting keypoints enables to identify any object appearing in the observed images. Popular keypoint extractors include classical non-parametric methods ( Lowe (1999), Bay et al. (2006)) as well as modern self-supervised learning approaches (Kulkarni et al. (2019), Gopalakrishnan et al. (2020)). However, keypoints from an image provide a representation based on the geometry of the field of view or, equivalently, of the pixel plane. This means that the intrinsic three-dimensional geometry of states of objects is not preserved since the representation differs from it by an unknown projective transformation. In specific situations such transformation can still be recovered by processing the extracted keypoints. This is the case when images are in first person view w.r.t. the observer: the keypoints can then be converted into three-dimensional landmarks via methods such as bundle adjustment (Triggs et al. (1999), Schonberger & Frahm (2016)).

Differently from computer vision approaches our framework is general, data-agnostic and does not rely on specific priors tied to the nature of observations. Assuming that the external object is interactable (in a particular sense), our idea is to extract geometric structure from performed actions based on functional properties alone (see Section 4.1).

**Active Perception.** The role of interaction in perception has been extensively studied in cognitive sciences and neuroscience (Held & Hein (1963); Gibson & Carmichael (1966); Noë et al. (2004)). Inspired by those, the field of active perception from robotics aims to enhance the understanding the world by part of an artificial systems via interaction (Bohg et al. (2017)). Applications include active control of cameras (Bajcsy (1988)) and manipulators (Tsikos & Bajcsy (1991)) in order to improve the perception of objects (Ilonen et al. (2014); Björkman et al. (2013); Schiebener et al. (2013)). Our work fits into the program of active perception since we rely on interaction in order to infer the representation. We show that actions enable to extract geometric and disentangled states from general and unstructured observations, which would not be possible without the the information gained from interaction.

## 3 FORMALISM AND ASSUMPTIONS

In this section we introduce the relevant mathematical formalism together with the assumptions necessary for our framework. We consider the following scenario: an agent navigates in a Euclidean space and interacts in an unknown way with an external object. This means that the space of states $\mathcal{S}$ is decomposed as

$$\mathcal{S} = \mathcal{S}_{\text{int}} \times \mathcal{S}_{\text{ext}} \tag{1}$$

where $\mathcal{S}_{\text{int}}$ is the space of states of the agent (*internal* states) and $\mathcal{S}_{\text{ext}}$ is the space of states of the object (*external* states). We identify both the agent and the object with their position in the ambient space, meaning that $\mathcal{S}_{\text{int}} \subseteq \mathbb{R}^n \supseteq \mathcal{S}_{\text{ext}}$. The actions that the agent performs are displacements of its state i.e., the space of actions consists of translations $\mathcal{A} = \mathbb{R}^n$. In our formalism we thus abstract objects as material points for simplicity. The practical extension of our framework to volumetric objects is discussed in Section 4.3 while the extension of agent's actions to arbitrary Lie groups is discussed Section 6.

Our first assumption (*connectedness*) is that the agent can reach any position from any other via a sequence of actions. This translates in the following topological condition:

**Assumption 3.1.** (Connectedness) *The space $\mathcal{S}_{\text{int}}$ is connected and open.*

When the agent performs an action $a \in \mathcal{A}$ the state $s = (s_{\text{int}}, s_{\text{ext}})$ transitions into a state denoted by $a \cdot s = (s'_{\text{int}}, s'_{\text{ext}})$. Since the actions displace the agent, the internal state gets translated as $s'_{\text{int}} = s_{\text{int}} + a$.[2] However, the law governing the transition of the object $s'_{\text{ext}} = T(s, a)$ is assumed to be unknown and can be arbitrarily complex and stochastic. We stick to deterministic transitions for simplicity of explanation. Crucially, the agent does not have access to the ground-truth state

---

[2]Whenever we write $a \cdot s$ we implicitly assume that the action is valid i.e., that $s_{\text{int}} + a \in \mathcal{S}_{\text{int}}$.

$s$. Instead it perceives unstructured and potentially high-dimensional observations $o \in \mathcal{O}$ (e.g., images) via an unknown emission map $\omega : \mathcal{S} \to \mathcal{O}$. We assume that $\omega$ is injectivite so that actions induce deterministic transitions of observations. The latter are denoted by abuse of notation as $o' = a \cdot o$. This assumption is equivalent to total observability of the scenario and again simplifies the forthcoming discussions by avoiding the the need to model stochasticity.

The fundamental assumption of this work is that the dynamics of the external object revolves around *contact* i.e., the object does not displace unless it is touched by the agent. This is natural and often satisfied in practice. In order to formalize it, note that when the agent in state $s_{\text{int}}$ performs an action $a \in \mathcal{A}$ we can imagine it moving along the open segment $\lfloor s_{\text{int}}, \ s_{\text{int}} + a \rfloor = \{s_{\text{int}} + ta\}_{0 < t < 1}$. Our assumption then translates into (see Figure 1 for a graphical depiction):

**Assumption 3.2.** (Interaction Occurs at Contact) *For all agent states $s_{\text{int}} \in S$ and actions $a \in \mathcal{A}$ it holds that $s'_{\text{ext}} = s_{\text{ext}}$ if and only if $s_{\text{ext}} \notin \lfloor s_{\text{int}}, \ s_{\text{int}} + a \rfloor$.*

As such, the dynamics of the external object can be summarized as follows:

$$s'_{\text{ext}} = \begin{cases} s_{\text{ext}} & \text{if } s_{\text{ext}} \notin \lfloor s_{\text{int}}, \ s_{\text{int}} + a \rfloor, \\ T(s, a) & \text{otherwise.} \end{cases} \tag{2}$$

Finally, we need to assume that interaction is possible for every state of the object i.e., the latter has to be always reachable by the agent. This is formalized via the following inclusion:

**Assumption 3.3.** (Reachability) *It holds that $\mathcal{S}_{\text{ext}} \subseteq \mathcal{S}_{\text{int}}$.*

## 4 METHOD

### 4.1 REPRESENTATIONS AND EQUIVARIANCE

We now outline the inference problem addressed in the present work. Given the setting introduced in Section 3, the overall goal is to infer a *representation* of observations

$$\varphi : \mathcal{O} \to \mathcal{Z} = \mathcal{Z}_{\text{int}} \times \mathcal{Z}_{\text{ext}} \tag{3}$$

where $\mathcal{Z}_{\text{int}} = \mathcal{Z}_{\text{ext}} = \mathbb{R}^n$. Ideally $\varphi$ recovers the underlying inaccessible state in $\mathcal{S} \subseteq \mathcal{Z}$ and disentangles $\mathcal{S}_{\text{int}}$ from $\mathcal{S}_{\text{ext}}$. In order to achieve this, our overall idea is to split the problem of representing the agent and the object. Since the actions of the agent are available $z_{\text{int}}$ can be inferred geometrically by existing methods from the literature. The representation of the object, $z_{\text{ext}} \in \mathcal{Z}_{\text{ext}}$, can then be inferred based on the one of the agent by exploiting the relation between the dynamics of the two (Equation 2). In order to represent the agent, we consider the fundamental concept of (translational) *equivariance*:

**Definition 1.** The representation $\varphi$ is said to be *equivariant* (on internal states) if for all $a \in \mathcal{A}$ and $o \in \mathcal{O}$ it holds that $z'_{\text{int}} = z_{\text{int}} + a$ where $(z_{\text{int}}, z_{\text{ext}}) = \varphi(o)$ and $(z'_{\text{int}}, z'_{\text{ext}}) = \varphi(a \cdot o)$.

We remark that Definition 1 refers to internal states only, making our terminology around equivariance unconventional. As observed in previous work (Marchetti et al. (2022)), equivariance guarantees a faithful representation of internal states. Indeed if $\varphi$ is equivariant then $z_{\text{int}}$ differs from $s_{\text{int}}$ by a constant vector. This means that the representation of internal states is a translation of ground-truth ones and as such is lossless (i.e., bijective) and isometrically recovers the geometry of $\mathcal{S}_{\text{int}}$.

The above principle can be leveraged in order to learn a representation of external states with the same benefits as the representation of internal ones. The intuition is that since the external object displaces only when it comes in contact with the agent (Assumption 3.2), $z_{\text{ext}}$ can be inferred by making it coincide with $z_{\text{int}}$. The following theoretical result formalizes the possibility of learning such representations and traces the foundation of our representation learning framework.

**Theorem 4.1.** *Suppose that the representation $\varphi : \mathcal{O} \to \mathcal{Z}$ satisfies:*

    *1. $\varphi$ is equivariant (Definition 1),*

    *2. $\varphi$ is injective,*

    *3. for all $o \in \mathcal{O}$ and $a \in \mathcal{A}$ it holds that $z'_{\text{ext}} \neq z_{\text{ext}}$ if and only if $z_{\text{ext}} \in \lfloor z_{\text{int}}, z_{\text{int}} + a \rfloor$ where $(z_{\text{int}}, z_{\text{ext}}) = \varphi(o)$ and $(z'_{\text{int}}, z'_{\text{ext}}) = \varphi(a \cdot o)$.*

*Then $\varphi \circ \omega$ is a translation i.e., there is a constant vector $h \in \mathbb{R}^n$ such that for all $s \in \mathbb{S}$ it holds that $\varphi(\omega(s)) = s + h$. In particular, $\varphi$ is an isometry w.r.t. the Euclidean metric on both $\mathbb{S}$ and $\mathbb{Z}$.*

We refer to the Appendix for a proof. Theorem 4.1 states that if the conditions $1. - 3.$ are satisfied (together with the assumptions stated in Section 3) then the representation recovers the inaccessible state up to a translation and thus faithfully preserves the geometry of the environment. All the conditions from 4.1 refer to properties of $\varphi$ depending on observations and the effect of actions on them, which are accessible and collectable in practice. The corresponding properties can be enforced to $\varphi$ by optimizing a loss, which will be discussed in the next section.

## 4.2 LEARNING THE REPRESENTATION

In this section we propose a representation learning framework implementing the conditions from Theorem 4.1 via a number of losses. We assume that the representation $\varphi = (\varphi_{\text{int}}, \varphi_{\text{ext}})$ consist of two parameterized functions $\varphi_{\text{int}} : \mathcal{O} \to \mathbb{Z}_{\text{int}}$, $\varphi_{\text{ext}} : \mathcal{O} \to \mathbb{Z}_{\text{ext}}$ e.g., two deep neural network models. In order to design the losses that the model(s) minimize w.r.t. the respective parameters, we consider a dataset of transitions $\mathcal{D} = \{(o, a, o' = a \cdot o)\}$ observed by the agent. Such $\mathcal{D}$ is collected by the agent exploring the environment and interacting with the external object as described in Section 3. We remark that our representation learning framework relies on supervision from interaction alone, without access to the ground-truth states of the external object nor of the agent itself.

First, we propose to enforce equivariance, condition 1 from Theorem 4.1, by minimizing the loss:

$$\mathcal{L}_{\text{int}}(o, a, o') = d(z'_{\text{int}}, z_{\text{int}} + a) \tag{4}$$

where $d$ is a measure of similarity on $\mathbb{Z}_{\text{int}} = \mathbb{R}^n$ and the notation is in accordance with Definition 1. Typically $d$ is chosen as the squared Euclidean distance. Equation 4 is a canonical loss which has been considered in previous work addressing equivariant representation learning (Mondal et al. (2020); Kipf et al. (2019)).

Next, we focus on condition 3. The idea is to force $\varphi_{\text{ext}}$ to minimize one of the following losses depending on the given datapoint:

$$\mathcal{L}_-(o, a, o') = d(z_{\text{ext}}, z'_{\text{ext}}) \qquad \mathcal{L}_+(o, a, o') = d(z_{\text{ext}}, \lfloor z_{\text{int}}, z_{\text{int}} + a \rfloor). \tag{5}$$

The distance involved in $\mathcal{L}_+$ represents a point-to-set metric and is typically set as $d(z, E) = \inf_{x \in E} d(z, x)$. The latter has a simple explicit expression in the case $E$ is a segment. One natural way to implement condition 3 is to combine the two losses from Equation 5 into a single one as:

$$\mathcal{L}_{\text{ext}}^0(o, a, o') = \min\{ \mathcal{L}_-(o, a, o'), \ \mathcal{L}_+(o, a, o') \}. \tag{6}$$

However, we have found that $\mathcal{L}_{\text{ext}}^0$ is hard to optimize in practice. This happens because $\mathcal{L}$ forces $\varphi$ to choose between $\mathcal{L}_-$ and $\mathcal{L}_+$ and, *at the same time*, to optimize the chosen one, resulting in an complex task. Instead, we rely on the following way of combining the two losses which leads to an easier optimization problem. We propose to deploy a latent *contrastive representation* in order to decide which one among $\mathcal{L}_-$ and $\mathcal{L}_+$ to minimize for a given datapoint. Namely we train another model $\varphi_{\text{cont}} : \mathcal{O} \to \mathcal{W}$ with latent space $\mathcal{W}$ (potentially different from $\mathbb{Z}$) which attracts $w = \varphi_{\text{cont}}(o)$ to $w' = \varphi_{\text{cont}}(o')$. Given a distance $d_{\mathcal{W}}$, we stick to the popular *InfoNCE* loss from contrastive learning literature (Chen et al. (2020)):

$$\mathcal{L}_{\text{cont}}(o, o') = d_{\mathcal{W}}(w, w') + \log \mathbb{E}_{o''} \left[ e^{-d_{\mathcal{W}}(w', w'') - d(z'_{\text{int}}, z''_{\text{int}})} \right] \tag{7}$$

where $o''$ is marginalized from $\mathcal{D}$. The second summand of Equation 7 encourages the joint encodings $(z_{\text{int}}, w)$ to spread apart. The latter avoids the collapse of $\varphi_{\text{cont}}$ to a constant and encourages condition 2 from Theorem 4.1 (injectivity). A typical choice is to normalize the output of $\varphi_{\text{cont}}$ so that $\mathcal{W}$ is a sphere and set $d_{\mathcal{W}}(w, w') = -\cos(\angle ww') = -w \cdot w'$ (Wang & Isola (2020)). The latent space $\mathcal{W}$ enables to choose whether to optimize $\mathcal{L}_-$ or $\mathcal{L}_+$ since subsequent observations where the agent does not touch the object will lay close in $\mathcal{W}$. We propose to partition (the given batch of) the dataset in two disjoint classes $\mathcal{D} = C_- \sqcup C_+$ by applying a natural thresholding algorithm to the quantities $d_{\mathcal{W}}(w, w')$. This can be achieved via one-dimensional 2-means clustering, which is equivalent to Otsu's algorithm (Otsu (1979)). We then optimize:

$$\mathcal{L}_{\text{ext}}(o, a, o') = \begin{cases} \mathcal{L}_-(o, a, o') & \text{if } (o, a, o') \in C_-, \\ \mathcal{L}_+(o, a, o') & \text{if } (o, a, o') \in C_+. \end{cases} \tag{8}$$

In summary, the total loss minimized by the models $(\varphi_{\text{int}}, \varphi_{\text{ext}}, \varphi_{\text{cont}})$ w.r.t. the respective parameters is (see the pseudocode included in the Appendix):

$$\mathcal{L} = \mathbb{E}_{(o,a,o') \sim \mathcal{D}}[\mathcal{L}_{\text{int}}(o, a, o') + \mathcal{L}_{\text{ext}}(o, a, o') + \mathcal{L}_{\text{cont}}(o, o')]. \tag{9}$$

### 4.3 Incorporating Shapes of Objects

So far we have considered the external object as a point in Euclidean space. However, the object typically manifests with a body and thus occupies a volume. Interaction and consequent displacement (Assumption 3.3) occurs when the agent comes in contact with the boundary of such volume. In order to represent states faithfully, the representation has to take the volume of the object into account. We propose to naturally achieve this via *stochastic* outputs i.e., by designing $z_{\text{ext}}$ as a probability density over $\mathcal{Z}_{\text{ext}}$ representing (a fuzzy approximation of) the body of the object. More concretely, the output of $\varphi_{\text{ext}}$ consists of (parameters of) a Gaussian distribution whose covariance matrix represents the body of the object. The distance $d$ appearing in Equation 5 is replaced with Kullback-Leibler divergence. The latter has an explicit simple expression for Gaussian densities which allows to compute $\mathcal{L}_-$ directly. In order to compute $\mathcal{L}_+$ we rely on a Monte Carlo approximation, meaning that we sample a point uniformly from the interval and set $\mathcal{L}^+$ as its negative log-likelihood w.r.t. the density of $z_{\text{ext}}$.

## 5 Experiments

We empirically investigate the performance of our framework in correctly identifying the position of an agent and of an interactive object. The overall goal of the experimental evaluation is showing that our representation is capable of extracting the geometry of states without relying on any prior knowledge of observations e.g., depth information. All the scenarios are normalized so that states lie in the unit cube. Observations are RGB images of resolution $100 \times 100$ in all the cases considered. We implement each of $\varphi_{\text{int}}$, $\varphi_{\text{ext}}$ and $\varphi_{\text{cont}}$ as a ResNet-18 (He et al. (2016)) and train them for 100 epochs via the Adam optimizer with learning rate 0.001 and batch-size 128.

### 5.1 Baselines and Evaluation Metrics

We compare our framework with two baselines:

- *Transporter Network* (Kulkarni et al. (2019)): a vision-based state-of-the-art unsupervised keypoint extractor. The approach heavily relies on image manipulation in order to infer regions of the pixel plane that are persistent between pairs of images. We train the model in order to extract two (normalized) keypoints representing $z_{\text{int}}$ and $z_{\text{ext}}$ respectively.

- *Variational AutoEncoder* (VAE) (Kingma & Welling (2013); Rezende et al. (2014)): a popular representation learner with a standard Gaussian prior on its latent space. We impose the prior on $\mathcal{Z}_{\text{ext}}$ only, while $\varphi_{\text{int}}$ is still trained via the equivariance loss (Equation 4). The decoder takes the joint latent space $\mathcal{Z}$ in input. We set $\dim(\mathcal{Z}_{\text{ext}}) = 32$. This makes the representations disentangled, so that $z_{\text{int}}$ and $z_{\text{ext}}$ are well-defined. The resulting representation of the object is generic and is not designed to extract any specific structure from observations.

In order to evaluate the preservation of geometry we rely on the following evaluation metric $\mathcal{L}_{\text{test}}$. Given a trained representation $\varphi : \mathcal{O} \to \mathcal{Z}$ and a test set $\mathcal{D}_{\text{test}}$ of observations with known ground-truth states, we define:

$$\mathcal{L}_{\text{test}} = \mathbb{E}_{o \sim \mathcal{D}_{\text{test}}} \left[ d(z_{\text{int}} - z_{\text{ext}}, s_{\text{int}} - s_{\text{ext}}) \right] \tag{10}$$

where $d$ is the squared Euclidean distance. Since both our framework and (the encoder of) VAE have stochastic outputs (see Section 4.3), we set $z_{\text{ext}}$ as the mean of the corresponding Gaussian distribution. Equation 10 measures the quality of preservation of the relative position between the agent and the object by part of the representation. When $\mathcal{L}_{\text{test}} = 0$, $\varphi$ is an isometry (w.r.t. the Euclidean metric) and thus recovers the geometry of states. The translational invariance of $\mathcal{L}_{\text{test}}$ makes the comparison agnostic to any reference frame eventually inferred by the given learner.

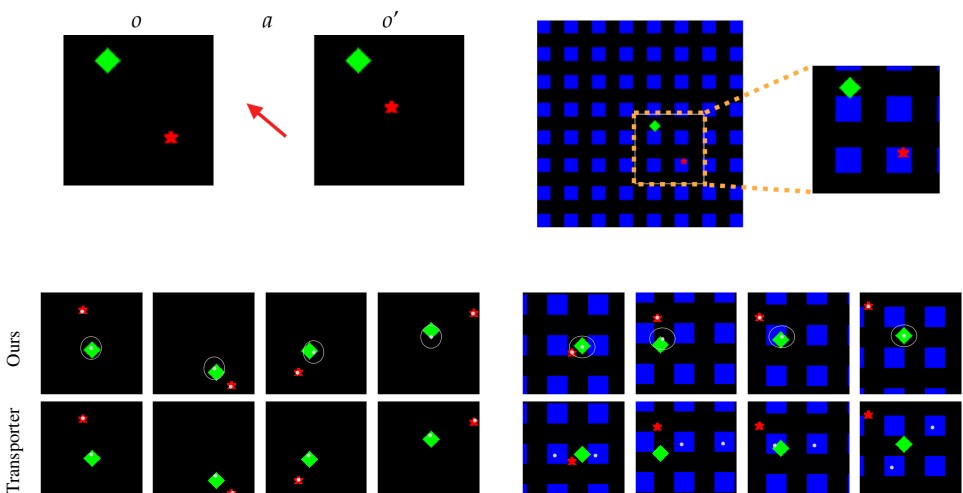

Figure 2: **Top:** Visualization of the dataset from the Sprites experiment. On the left, an example of an observation $o$, an action $a$ and the resulting observation $o'$. On the right, an example of an observation from the second version of the dataset where a moving background is added as a visual distractor. **Bottom:** Comparison of $z_{\text{int}}$, $z_{\text{ext}}$ (gray dots, with the ellipse representing the learned std) extracted via our model and the Transporter network on both versions of the Sprites dataset.

## 5.2 SPRITES

For the first experiment we procedurally generate images of two sprites (the agent and the object) moving on a black background (see Figure 2). Between images, the agent (red figure) moves according to a known action. If the agent comes in contact with the object (green diamond) during the execution of the action (see Assumption 3.2) the object is randomly displaced on the next image. In other words, the object's transition function $T(s, a)$ is stochastic with a uniform distribution. Such completely stochastic dynamics highlights the independence of the displacement of the agent w.r.t. the one of the object. We additionally consider a second version of the experiment by simulating a moving background. Images are now overlayed on top of a three-times larger second image (blue squares in Figure 2, right). The field of view and thus the background moves together with the agent. The moving background behaves as a visual distractor and makes it challenging to extract structure (e.g., keypoints) via computer vision.

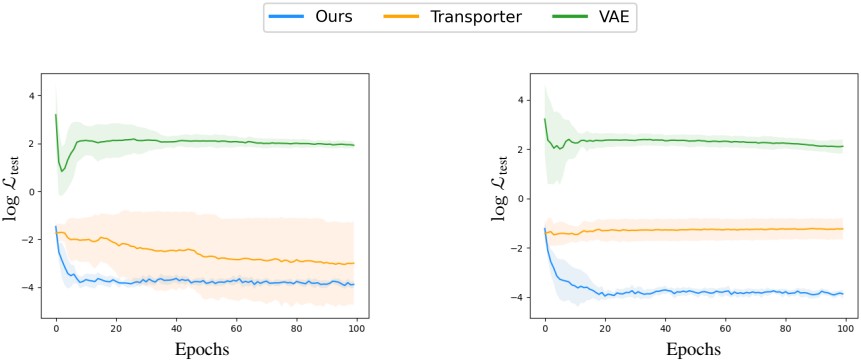

Figure 3: Log-scale plots of the evaluation metric (Equation 10) as the training progresses for the Sprite experiment. The curves display mean and std (for 10 experimental runs). **Left**: standard version of the dataset. **Right:** challenging version with moving background.

Figure 3 displays the analytic comparison of the performances between our model and the baselines in terms of the evaluation metric (Equation 10). The plot is in log-scale for visualization purposes. Moreover, Figure 2 (bottom) reports a qualitative comparison between our model and the Transporter network. As can be seen, for the simpler version of the experiment (plot on the left) both our model and the Transporter network successfully achieve low error and recover the geometry of both the agent and the object. Note that the Transporter network converges slowly and with high variance (Figure 3, left). This is probably due to the presence of a decoder in its architecture. Our framework instead involves losses designed directly in the latent space, avoiding an additional model to decode observations. As expected, VAE achieves significantly worse performances because of the lack of structure in its representation.

For the more challenging version of the experiment, the transporter is not able to extract the expected keypoints. The moving background causes the model to focus on regions of the image not corresponding to the agent and the object. This is reflected by a significantly higher error (and variance) w.r.t. our framework (Figure 3, right). The latter still infers the correct representation and preserves geometry. This empirically confirms that our model is robust to visual distractors since it does not rely on any data-specific feature or structure.

### 5.3 Soccer

For the second experiment we test our framework on an environment consisting of an agent on a soccer field colliding with a ball (see Figure 4, top-left). The scene is generated and rendered via the Unity engine. The physics of the ball is simulated realistically: in case of contact, rolling takes gravity and friction into account. Observations are third-person views of the scene from a fixed camera. Note that even though the scene is generated via three-dimensional rendering, the (inaccessible) state space is still two-dimensional since the agent navigates on the field. Our dataset consists of 10000 generated triples $(o, a, o' = a \cdot o)$.

Figure 4 (top-right) displays the comparison of the performances between our model and the baselines in terms of the evaluation metric (Equation 10). On the bottom-left, we report visualizations of the learned representations. The Transporter network correctly extracts the keypoints on the *pixel plane*. As discussed in Section 2, such a plane differs from $S_{\text{int}}$ by an unknown projective (and thus non-isometric) transformation. This means that despite the successful keypoint extraction, the geometry of the state space is not preserved, which is reflected by the high error on the plot. This is a general limitation of vision-based approaches: they are unable to recover the intrinsic geometry due to perspective in case of a three-dimensional scene. Differently from that, our framework correctly represents the scene and achieves low error independently from observations.

### 5.4 Control Task

In our last experiment we showcase the benefits of our representations in solving downstream control tasks. The motivation is that a geometric and low-dimensional representation improves efficiency and generalization compared to solving the task directly from observations. To this end we design a control task for the Soccer environment consisting in kicking the ball *into the goal*. The reward is given by the negative distance between the (barycenter of the) the ball and the (barycenter of the) goal. In each episode the agent and the ball is initially placed in a random location while the ball is placed in the center. The maximum episode length is 20 steps. We then train a number models via the popular reinforcement learning method *Proximal Policy Optimization* (PPO; Schulman et al. (2017)). One model (*End-to-End*) receives raw observations as inputs. The others operate on pretrained representations $\mathcal{Z}$ given by the Transporter network, the VAE and our method respectively. All the models implement a comparable architecture for a fair comparison.

Figure 4 (bottom-right) displays the reward gained on test episodic runs as the training by reinforcement learning progresses. As can be seen, our geometric representation enables to solve the task more efficiently than both the competing representations (Transporter and VAE) and the end-to-end model. Note that the Transporter not only does not preserve the geometry of the state space, but has the additional disadvantage that the keypoint corresponding to the agent and the object can get swapped in the output of $\varphi$. This causes indeterminacy in the representation and has a negative impact on solving the task. Due to this, the Transporter performs similarly to the end-to-end model and is outperformed by the generic and non-geometric representation given by the VAE. In conclusion,

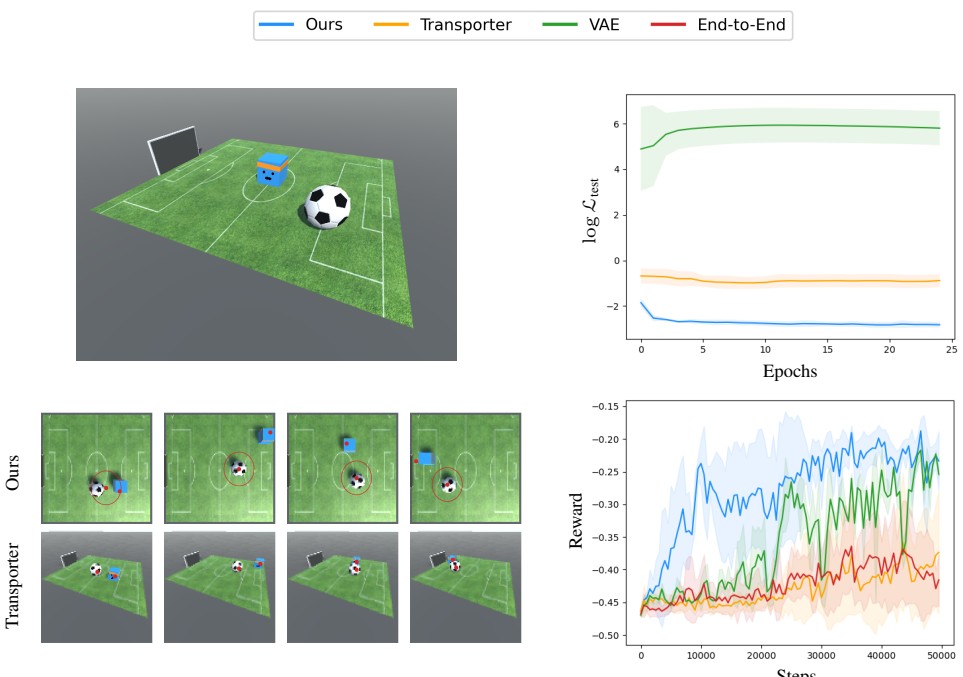

Figure 4: **Top:** on the left, an example of an observation from the Soccer experiment. On the right, log-scale plot of the evaluation metric as the training progresses. **Bottom:** on the left, visual comparison of $z_{\text{int}}$, $z_{\text{ext}}$ (red dots) extracted via our model and the Transporter network. For our model, we visualize a view of the scene from the top instead of the original observation. On the right, plot of the reward gained via reinforcement learning on top of different representations.

the results show that a downstream learner can significantly benefit from geometric representations of observations in order to solve downstream control tasks.

# 6 CONCLUSIONS AND FUTURE WORK

In this work we proposed a novel framework for learning representations of both an agent and an object the agent interacts with. We designed a system of losses based on a theoretical principle that guarantees isometric representations independently from the nature of observations and relying on supervision from performed actions alone. We empirically investigated our framework on multiple scenarios showcasing advantages over computer vision approaches.

Throughout the work we assumed that the agent interacts with a single object. An interesting line of future investigation is extending the framework to take multiple objects into account. In the stochastic context (see Section 4.3) an option is to model $z_{\text{ext}}$ via multi-modal densities, with each mode corresponding to an object. This however comes with a computational challenge: since the dataset has to cover transitions densely for a successful inference, the sample complexity would grow exponentially w.r.t. the number of objects. An improvement would thus be necessary in order to design a sample-efficient method. Similarly, another possible extension of our model is considering multiple agents simultaneously interacting with external objects.

As an additional line for future investigation, our framework can be extended to actions beyond translations in Euclidean space. Lie groups other than $\mathbb{R}^n$ often arise in practice. For example, if the agent is able to rotate its body then (a factor of) the space of actions has to contain the group of rotations $\text{SO}(n)$, $n = 2, 3$. Thus, a framework where actions (and consequently states) are represented in general Lie groups defines a useful and interesting extension.

## 7 REPRODUCIBILITY STATEMENT

We include Python code for all the experiments and baselines in the supplementary material. For the Sprites experiment, we include the code for generating data procedurally. For the Soccer experiment, we include an anonymized link to download the dataset and code to run the reinforcement learning models in Unity. This makes our experiments fully reproducible. For the main theoretical result (Theorem 4.1), the complete proof is included in the Appendix while all the assumptions are stated in Section 3.

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

## A APPENDIX

### A.1 PROOFS OF THEORETICAL RESULTS

**Theorem A.1.** *Suppose that the representation $\varphi : \mathcal{O} \to \mathcal{Z}$ satisfies:*

1. *$\varphi$ is equivariant (Definition 1),*

2. *$\varphi$ is injective,*

3. *for all $o \in \mathcal{O}$ and $a \in \mathcal{A}$ it holds that $z'_{\text{ext}} \neq z_{\text{ext}}$ if and only if $z_{\text{ext}} \in \lfloor z_{\text{int}}, z_{\text{int}} + a \rfloor$ where $(z_{\text{int}}, z_{\text{ext}}) = \varphi(o)$ and $(z'_{\text{int}}, z'_{\text{ext}}) = \varphi(a \cdot o)$.*

*Then $\varphi \circ \omega$ is a translation i.e., there is a constant vector $h \in \mathbb{R}^n$ such that for all $s \in \mathcal{S}$ it holds that $\varphi(\omega(s)) = s + h$. In particular, $\varphi$ is an isometry w.r.t. the Euclidean metric on both $\mathcal{S}$ and $\mathcal{Z}$.*

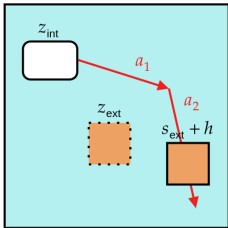 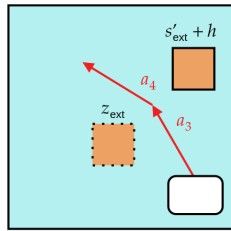

Figure 5: Graphical depiction of the proof of Theorem 4.1.

*Proof.* Pick an arbitrary state $s^0 \in \mathcal{S}$ together with its represented internal state $z_{\text{int}}^0$ and set $h = z_{\text{int}}^0 - s_{\text{int}}^0$. For any state $s$, consider the action $a = s_{\text{int}} - s_{\text{int}}^0$. Equivariance then implies that $z_{\text{int}} = z_{\text{int}}^0 + a = s_{\text{int}} + h$. This shows that the claim holds for internal states.

To prove that the same happens for external states, suppose by contradiction that there is a state $s$ such that $z_{\text{ext}} \neq s_{\text{ext}} + h$. Consider any path during which the agent interacts with the object without passing through $z_{\text{ext}}$. Formally, this means considering a sequence of actions $a_1, \cdots, a_r$ such that (see Figure 5, left):

- $z_{\text{ext}}$ and $s_{\text{ext}} + h$ do not belong to $\lfloor z_{\text{int}} + a_1 + \cdots + a_{i-1}, z_{\text{int}} + a_1 + \cdots + a_i \rfloor$ for every $i = 1, \cdots, r - 1$,

- $z_{\text{ext}}$ does not belong to $\lfloor z_{\text{int}} + a_1 + \cdots + a_{r-1}, z_{\text{int}} + a_1 + \cdots + a_r \rfloor$ but $s_{\text{ext}} + h$ does.

The existence of such a path follows from Assumptions 3.1 and 3.3. After interaction the state becomes $s' = a_r \cdot (a_{r-1} \cdots (a_1 \cdot s))$ with $s'_{\text{ext}} \neq s_{\text{ext}}$ because of Assumption 3.2. One can then consider a path back to the initial agent's position $z_{\text{int}}$ i.e., another sequence of actions $a_{r+1}, \cdots, a_R$ such that (see Figure 5, right):

- $s'_{\text{ext}} + h$ and $z_{\text{ext}}$ do not belong to $\lfloor z_{\text{int}} + a_1 + \cdots + a_{i-1}, z_{\text{int}} + a_1 + \cdots + a_i \rfloor$ for every $i = r + 1, \cdots, R$,

- $a_1 + \cdots + a_R = 0$.

All the conditions imply together that the representation of the object remains equal to $z_{\text{ext}}$ during the execution of the actions $a_1, \cdots, a_R$. Since the actions sum to $0$, the representation of the agent does not change as well. But then $\varphi(\omega(s)) = \varphi(\omega(s_{\text{int}}, s'_{\text{ext}}))$ while $s_{\text{ext}} \neq s'_{\text{ext}}$, contraddicting injectivity. We conclude that $z_{\text{ext}} = s_{\text{ext}} + h$ and thus $z = s + h$ as desired. $\qquad\square$

## A.2 PSEUDOCODE FOR LOSS COMPUTATION

---

**Algorithm 1** Loss Computation

---

**Input:** Batch $\mathcal{B} \subseteq \mathcal{D}$, models $\varphi_{\text{int}}, \varphi_{\text{ext}}, \varphi_{\text{cont}}$
**Output:** Loss $\mathcal{L}$

  $\mathcal{L} = 0$
  **for all** $(o, a, o') \in \mathcal{B}$ **do**
    Compute $z_{\text{int}} = \varphi_{\text{int}}(o), z_{\text{ext}} = \varphi_{\text{ext}}(o), z'_{\text{int}} = \varphi_{\text{int}}(o'), z'_{\text{ext}} = \varphi_{\text{ext}}(o'), w = \varphi_{\text{cont}}(o), w' = \varphi_{\text{cont}}(o')$
  **end for**
  Compute the classes $C_-, C_+$ via Otsu's algorithm based on $\{d_\mathcal{W}(w, w')\}$
  **for all** $(o, a, o') \in \mathcal{B}$ **do**
    Compute $\mathcal{L}_{\text{int}}(o, a, o')$ via Equation 4
    Compute $A = \{d_\mathcal{W}(w', w''), \ d(z'_{\text{int}}, z''_{\text{int}})\}$ **for** $o''$ marginalized from $\mathcal{B}$
    Based on $A$ compute $\mathcal{L}_{\text{cont}}(o, o')$ via Equation 7
    **if** $d_\mathcal{W}(w, w') \in C_-$ **then**
      Compute $\mathcal{L}_{\text{ext}}(o, a, o') = \mathcal{L}_-(o, a, o')$ via Equation 5 (left)
    **else**
      Compute $\mathcal{L}_{\text{ext}}(o, a, o') = \mathcal{L}_+(o, a, o')$ via Equation 5 (right)
    **end if**
    $\mathcal{L} \leftarrow \mathcal{L} + \mathcal{L}_{\text{int}} + \mathcal{L}_{\text{ext}} + \mathcal{L}_{\text{cont}}$
  **end for**

---

