# OpenReview forum: "Learning Geometric Representations of Interactive Objects"
_ICLR.cc/2023/Conference — Submitted to ICLR 2023_

### Official Review · Reviewer_a3YM · 2022-10-21

**Confidence:** 3
**Correctness:** 3
**Technical Novelty And Significance:** 2
**Empirical Novelty And Significance:** 1
**Recommendation:** 3

**Clarity, Quality, Novelty And Reproducibility:**

The paper is clearly written and results / derivations are well presented. The python code for reproducing the results is included in supplementary materials. To the reviewer's best knowledge, the paper sufficiently cite relevant papers in this domain, even though the reviewer is not up to date with the most recent publications.

**Strength And Weaknesses:**

Strength:
Geometry-aware representations are of great importance in modeling agent-environment interaction, with rich potential applications. This paper focuses on an interesting scenario of learning Euclidean positions of point-like agent and environmental objects. Experiments successfully demonstrates the geometry awareness of the learned representation, as well as its application in simple control tasks.

Weaknesses:
1. The proposed formulation seems to require the agent state and environment state to be embedded in the same geometric space. It is unclear how the method generalizes to states without a shared geometric grounding between them, such as object properties.

2. The logic of the paper is not very clearly laid out, especially the relation between the theoretical derivations and empirical studies. The theory part does not take the discussion much further than the introduction of the objective functions. The paper could have been a lot stronger if more contents are dedicated to empirical studies.

3. Experimental setup are too simplified (partly due to the fundamental limitation of the formulation). The paper makes claim about generality of its proposal hence it is expected to see the method plays out under more realistic environments and weaker assumptions (ideally more complex than point-objects). Two simplified environments x two baselines seem insufficient for this purpose.

**Summary Of The Paper:**

This paper proposes a framework for learning geometric-aware disentangled representations via agent-environment interaction. This goal is achieved by introducing an equivariant loss term on top of the ordinary distance losses to enforce geometric disentanglement between agent and environment state, together with a contrastive loss to assist learning. The paper provides detailed mathematical formulation of the general setup and conducts empirical studies on simple datasets whose state space are Euclidean. Experiments show preliminary comparisons between the proposed method and two other self-supervised representation learning baselines.

**Summary Of The Review:**

Even though this paper conducts a close-loop study of a novel representation learning framework, I find it lacking justifications to its limitations as well as empirical studies. A lot of work is yet to be done for this paper to reach its true potential. My current assessment is reject.

---

> ### Author Response · Authors · 2022-11-10
> **Reply to Reviewer a3YM**
>
> We wish to thank the reviewer for the comments and the constructive criticism. We would like to comment on some of the points raised in the review.
>
> **It is unclear how the method generalizes to states without a shared geometric grounding between them, such as object properties.**
>
> Our work addresses the problem of extracting only the geometric position of the agent and the object. The applications we have in mind are control tasks where the object has to be displaced by the agent in some way, such as the Soccer experiment we present in Section 5.4. Thus, we would like our representation to be \emph{invariant} to the additional features (e.g., color, mass etc.) instead of accounting for them since such properties are irrelevant for the task.
>
> Moreover, since we assume that the only form of self-supervision consists of the performed actions (as standard in control problems), features that are not influenced by the actions (e.g., color, mass) are unfeasible  to infer geometrically in the general setting of our work. If one wants to still retain the information of such features in the representation, an option is to deploy our framework together with a generic and non-geometric representation aiming for losslessness such as a (variational) autoencoder.
>
> **The logic of the paper is not very clearly laid out, especially the relation between the theoretical derivations and empirical studies. The theory part does not take the discussion much further than the introduction of the objective functions.**
>
> We wish to expand on the significance of the theoretical discussion of our work as well as its connection to the experimental evaluation. Our main theoretical contribution is Theorem 4.1., which shows that it is possible to infer the pose of both the agent and the object geometrically (i.e. via an isometry) based on self-supervision from actions alone. This traces the foundation of our framework: all the losses described in Section 4.2 are designed to encourage conditions 1), 2) and 3) from Theorem 4.1, and the latter thus guarantees that at the global minimum the model infers isometric representations.
>
> The experimental investigation is related to the above in that the evaluation metric we deploy (Equation 10) measures how close the representation $\varphi$ is to an isometry.  The fact that this metric reaches values close to 0 (see the log-scale plots in Figure 3 and top-right of Figure 4) empirically confirms that the model learns the isometric representation as expected from Theorem 4.1. This is also confirmed qualitatively by the visualizations of the representation (bottom of Figure 2 and bottom-left of Figure 3), where the learned points can be seen to visually overlap with the ground-truth positions of the object and agent respectively.
>
> **Experimental setup are too simplified (partly due to the fundamental limitation of the formulation). The paper makes claim about generality of its proposal hence it is expected to see the method plays out under more realistic environments and weaker assumptions (ideally more complex than point-objects)**
>
> We wish to remark that our framework actually goes beyond ‘point-like’ objects by taking into account volumetric objects together with their shape and pose. This is discussed in Section 4.3 and implemented in the model by default: $\varphi_\rm{ext}$ outputs Gaussians whose covariance (i.e., the inertia ellipsoid) represents the shape and orientation of the object. In fact, all the experiments considered involve objects that occupy volume in space. The learned shapes are visualized as ellipses in Figure 2 and Figure 3. The theoretical framework from Sections 3 and 4 defines the object and the agent as ‘point-wise’ for simplicity of explanation, but the theory holds in the volumetric setting with assumptions analogous to the ones in Section 3. We agree that this is unclear in our manuscript. We have rephrased parts of Section 4.3 and emphasized it in Section 3 in the updated version.

---

### Official Review · Reviewer_tBxf · 2022-10-22

**Confidence:** 3
**Correctness:** 3
**Technical Novelty And Significance:** 2
**Empirical Novelty And Significance:** 2
**Recommendation:** 5

**Clarity, Quality, Novelty And Reproducibility:**

  - Not sure why the caption of figure 4 says "on both versions of the Sprites dataset".
  - Sec 5.3, why the inaccessible state space is 2D?
  - Since the paper only focus on Euclidean state space, why does figure 1 characterize the state space as a saddle surface?
  - How is the action defined? Is it simply the offset of the agent?
  - Doesn't the Soccer environment violate the assumption that the state of object remains constant when not acted upon? Why the training objective still makes sense?
  - The visualization shows that the learned visualization accurately capture the true state. How is the translation ambiguity is handled?


**Strength And Weaknesses:**

  - Strength
    - The experiments on the Soccer environment show that the proposed method can extract plausible representations under projected observations.
    - The experiments on the control task demonstrate the quality of the inferred agent and object states.
  - Weakness
    - Interaction between objects is limited to simple collisions. Thus, it is not clear to me how general this framework is.
    - It is not clear to me the precise meaning of "geometric representation". The paper mainly focuses on the location of the agent and object.
    In this case, both the agent and object can be abstracted as a point (with no meaningful rotation). I didn't find any discussion on agent or object shapes.
    - The true state dimension and the form of action (additive in this case) are assumed to be known.
    - The experiments are overall weak.
      - Transporter Network by design does not consider any action, thus, it cannot filter out distractors. But I do believe if the author could visualize all key points, the agent and the object should be well captured. A fair comparison would be to apply post-processing to the set of key points and pick out the top two that are most correlated with actions.
      - For the control task, judging by the reward plot, the training does not converge yet.

**Summary Of The Paper:**

  This paper follow the equivariant representation learning literature and proposed a equivariant representation learning framework for one-agent-one-object environment.
  By assuming that action additively change the state of the agent and the state of the object can only be changed by agent upon impact, the proposed model managed to recovery the underlining state space.


**Summary Of The Review:**

  The paper in its current state demonstrates the potential of this approach but did not fully justify it.
  I think many assumption/limitation of this approach requires in-depth discussion.

---

> ### Author Response · Authors · 2022-11-10
> **Reply to Reviewer tBxf (Part 1)**
>
> We thank the reviewer for the detailed comments, questions and points raised. Below we wish to comment and answer to the latter.
>
> **both the agent and object can be abstracted as a point (with no meaningful rotation). I didn't find any discussion on agent or object shapes.both the agent and object can be abstracted as a point (with no meaningful rotation). I didn't find any discussion on agent or object shapes.**
>
> We agree with the reviewer that considering shapes and orientations is important. In our theoretical discussion the object is abstracted as points for simplicity of explanation. However, in Section 4.3 we discuss how to incorporate volumetric objects (together with their pose) in the framework. This is done by outputting a Gaussian whose covariance (i.e., its inertia ellipsoid) represents the shape of the object together with its orientation. This is implemented in our model and in our experiments  we visualize the learned covariance as an ellipse (see Figure 2 and Figure 3). We agree that this is not clear enough in the manuscript. In the updated version we have rephrased Section 4.3 and emphasized it in Section 3.
>
> As for the agent, incorporating poses (i.e., rotations) is briefly discussed in Section 6. To this end, one needs to model the intrinsic state space of the agent as a \emph{Lie group} beyond translations e.g., the group of rotations $\rm{SO}(n)$. All the theoretical results and methods extend in principle to arbitrary Lie groups with assumptions analogous to the ones in Section 3 and 4. Although interesting and useful, we believe that the extension of our framework to Lie groups falls outside of the scope of the present work and we leave it for future investigation.
>
> **But I do believe if the author could visualize all key points, the agent and the object should be well captured. A fair comparison would be to apply post-processing to the set of key points and pick out the top two that are most correlated with actions.**
>
> We believe this is an interesting idea. However, we think that processing the keypoints statistically will hardly lead to isolating the agent and the object. This is because the background distractors in our Sprite experiment move together with the agent (see top-right of Figure 2), which is a realistic phenomenon. Thus, correlating keypoints with actions will not lead to distinguishing a keypoint corresponding to the agent from one corresponding to a distractor. Generally speaking, we believe that such a task is challenging since it is an example of statistical disentanglement, which is known to be subtle (see [1]).
>
> [1] Locatello et al., Challenging Common Assumptions in the Unsupervised Learning of Disentangled Representations, ICML 2019.
>
> **For the control task, judging by the reward plot, the training does not converge yet**
>
> We agree with the reviewer that some methods (e.g., End-to-End) have not converged in the plot. However, our intention in Section 5.4 is to show that geometrically structured representations improve the efficiency of reinforcement learning, not that they achieve higher rewards. Indeed, a powerful enough reinforcement learner will eventually solve the task on any lossless representation. What we conclude from the plot is that due to their structure, our representations enable the reinforcement learning algorithm to converge to the solution significantly faster than unstructured representations (VAE) and the original representation (End-to-End).

---

> ### Author Response · Authors · 2022-11-10
> **Reply to Reviewer tBxf (Part 2)**
>
> **Not sure why the caption of figure 4 says "on both versions of the Sprites dataset".**
>
> This is a typo. We thank the reviewer for noticing it and we have corrected it in the updated version.
>
> **Sec 5.3, why the inaccessible state space is 2D?**
>
> Even though the scene is rendered in 3D, the vertical coordinate is fixed for both the agent and the object since they translate on the field. Thus, the intrinsic state space is two-dimensional and corresponds to the field.
>
> **Since the paper only focus on Euclidean state space, why does figure 1 characterize the state space as a saddle surface?**
>
> In our intentions, Figure 1 is meant to represent the state space embedded in the observation space (via the emission map). While the state space is indeed Euclidean as pointed out by the reviewer, it gets deformed in the unstructured observation space (e.g., the space of images). Our framework recovers the state space (up to isometry) from the observation space. This is the message we aim to convey by drawing the embedded state space as a curved manifold in Figure 1.
>
> **How is the action defined? Is it simply the offset of the agent?**
>
> Yes, the action is the vector by which the agent displaces. In other words, the agent’s position transitions by translation in the state space. This is introduced formally in the second paragraph of Section 3.
>
> **Doesn't the Soccer environment violate the assumption that the state of object remains constant when not acted upon? Why the training objective still makes sense?**
>
> In order to not violate the assumption in the Soccer environment, after interacting with the ball the agent waits until the ball reaches equilibrium before recording an observation and performing the next action. This way the object’s state remains indeed invariant between two successive observations if no interaction occurs.
>
> **The visualization shows that the learned visualization accurately capture the true state. How is the translation ambiguity is handled?**
>
> In order to handle the translational ambiguity in the visualization, we align the learned representation by computing the minimum and the maximum value obtained in each dimension by $z_\rm{int}$ on the training data. The resulting rectangle is overlapped with the image.

---

### Official Review · Reviewer_ySVW · 2022-10-25

**Confidence:** 3
**Correctness:** 3
**Technical Novelty And Significance:** 3
**Empirical Novelty And Significance:** 3
**Recommendation:** 6

**Clarity, Quality, Novelty And Reproducibility:**

The paper clarity was generally good (barring a few points raised in weaknesses). The work appears to be novel and reproducible.

**Strength And Weaknesses:**

# Strengths

* The idea is well motivated and clearly described. In particular, Sections 3 and 4 lay the foundation for inexperienced readers and have good clarity. The assumptions are also clearly stated.
* I have checked the theoretical foundation and proof (to the best of my ability) and they are technically correct.
* The code has been provided for reproducibility.
* The experiments, while toyish, are well designed and show consistent improvements over baselines. The qualitative analysis from Figures 2 and 4 are helpful to understand the model in action.

# Weaknesses

## Some assumptions are unjustified and could be limiting
* Page 4, line 3: "We assume injectivity in the emission …" --- doesn't this exclude any form of partial observability (for example, the object may not be visible, or the agent could be facing a blank wall in with limited field of view)?
* Theorem 4.1, condition 3 - the "if and only if" excludes cases where the agent chooses not to interact with the object (i.e., the agent moves past the object without affecting it).
* Sec. 4.2, lines 5-6: "Such D is collected by the agent exploring the environment …" --- this assumes that there are sufficient examples of the agent interacting with the object. How is this guaranteed while generating the training triplets?

## Unclear explanation of Equation 7
* I was not able to fully grasp how L_cont is derived in Eqn. 7. It is also unclear what the latent-space W is capturing, and how this helps select between L- and L+.

## Evaluation metric entangles inference of z_int and z_ext
* Is it possible to independently evaluate z_int and z_ext?
* For example, we could compare (z_int@t - z_int@0) with (s_int@t - s_int@0), and (z_ext@t - z_ext@0) with (s_ext@t - s_ext@0).
* It is unclear how much of an error exists in inferring agent state vs. object state.

## VAE converges to comparable RL performance while being very poor at inferring states
* In Fig 4  - VAE performs very poorly when it comes to inferring the agent and object states, but it eventually achieves comparable RL task reward when compared to the proposed method.
* Does this imply that we may not need isometric representations that are translationally shifted from the agent / object states? Perhaps less structured representations are sufficient.

## One point not clear in proof of theorem A1
* Page 12 - "All the conditions imply ... remains equal to z_ext during the execution of actions a1 ... aR" -- why is this the case? After interaction, the object representation could have changed to some z_ext^’ != s_ext^’ + h right?



**Summary Of The Paper:**

This paper proposes a representation learning framework for identifying agent and object representations from observations in the scenario of an agent interacting with an object. In particular, the work aims to learn an isometric representation (i.e., capture underlying geometric states in a loss-less fashion) with object and agent states disentangled. The work also lays the theoretical foundation for this framework and formally proves that an ideal learner can successfully learn such representations. Experiments are presented on grid-worlds and a soccer environment.

**Summary Of The Review:**

The idea is interesting, well-motivated and clearly described. The experiments are well designed and demonstrate the fundamental points shown in the theoretical framework. I find that some assumptions may be strong and limiting, and a few other weaknesses in terms of clarity of Eqn. 7, the evaluation metric, and inconsistency b/w VAE representation performance vs. RL performance. I would appreciate it if the authors could address these concerns.

---

> ### Author Response · Authors · 2022-11-10
> **Reply to Reviewer ySVW (Part 1)**
>
> We thank the reviewer for the thorough review and for the relevant point raised, on which we wish to comment.
>
> **Doesn't this exclude any form of partial observability (for example, the object may not be visible, or the agent could be facing a blank wall in with limited field of view)?**
>
> The reviewer is correct: injectivity of the emission map is equivalent to total observability. We make this assumption in order to simplify the theoretical discussion since in the context of partial observability the transition of observations is stochastic. For example, if the agent faces a blank wall then its position is undetermined by $o$ and thus $a \cdot o$ is distributed according to the possible observations after performing $a$. We have briefly expanded on this in the updated version.
>
> Our framework can in the future be adapted to the partially observable setting by standard techniques from control and reinforcement learning. One option is to implement $\varphi$ as a sequence model and input sequences of observations in order to make observations more informative. Another option is to rely on probabilistic modeling i.e., incorporating additional stochasticity in the representation in order to model the uncertainty around the position.
>
> **Theorem 4.1, condition 3 - the "if and only if" excludes cases where the agent chooses not to interact with the object**
>
> Indeed the assumption here is that the agent is not allowed to perform an action colliding with the object (i.e., $z_\rm{ext} belonging to the interval $\z_\rm{int}, z_\rm{int} + a$) without displacing it (i.e., changing $z_\rm{ext}$). This assumption is necessary for the proof of Theorem 4.1 and is the analogue in the representation of Assumption 3.2 on the ground-truth dynamics. The reason for those assumptions is that otherwise there might not be enough interaction between the agent and the object in order to infer the object’s position. In practice, the assumption can be weakened statistically by allowing the agent to touch the object without displacing it. As long as enough displacements are observed, our learning framework would still be applicable. This is however hard to formalize and to study theoretically, and we prefer to formally stick to an ‘ideal’ assumption, albeit slightly more restrictive.
>
> **"Such D is collected by the agent exploring the environment …" --- this assumes that there are sufficient examples of the agent interacting with the object. How is this guaranteed while generating the training triplets?**
>
> It is true that enough interactions need to be observed in order to learn successfully. The way we generate the triplets is as follows: we choose a Bernoulli parameter $p \in [0,1]$ representing the frequency of interaction. Then a proportion $p$ of the dataset is generated as a triplet where interaction happens, with the agent and the objects spawning at a random location and performing a random action that touches the object. In the experiments in the paper, $p$ is set to $0.5$ i.e., half of the data involves interaction. The simulator for the Sprites experiments included in our provided code allows to change the parameter $p$. We have experimented with  $p$ as low as $0.1$ and have not seen any meaningful change in performance. We conclude that the amount of interaction that needs to be observed is low in practice and that the model is robust w.r.t. $p$.
>
> **I was not able to fully grasp how L_cont is derived in Eqn. 7. It is also unclear what the latent-space W is capturing, and how this helps select between L- and L+.**
>
> Equation 7 comes from the standard theory of contrastive learning. It forces $\varphi_\rm{cont}$ to be as invariant to actions as possible (first summand of Equation 7) while encouraging injectivity for $(\varphi_\rm{cont}, \varphi_\rm{int})$ (second summand). This means that $\mathcal{W}$ will capture the information that is most invariant to actions while still preserving all the variability of data that is complementary to $z_\rm{int}$ (the agent’s position). Thus, embeddings to $\mathcal{W}$ will correspond to the object’s position, but in an unstructured and non-geometric way. This means that even though the geometric position of the object can not be extracted directly from $\mathcal{W}$, the latter can be used to determine whether the object has been displaced between $o$ and $o’$ by looking at how close  $w$ and $w’$ are. This is how we use $\mathcal{W}$: by looking at the statistics of the distances $d_\mathcal{W}(w,w’)$ (via the Ostu’s algorithm) our framework determines whether interaction has happened or not and decides which loss to optimize accordingly.

---

> ### Author Response · Authors · 2022-11-10
> **Reply to Reviewer ySVW (Part 2)**
>
> **Is it possible to independently evaluate z_int and z_ext?**
>
> We thank the reviewer for the suggestion and we believe it raises an interesting point. We have computed the proposed scores (w the squared Euclidean distance and where the base point $s_0$ is chosen randomly) on the Sprite experiment (in its simple version) at convergence (100 epochs). The score is $4.7 \times 10^{-5}$ (std $5.8 \times 10^{-5}$) for the agent while it is $6.7 \times 10^{-3}$ (std $6.0 \times 10^{-3}$). The two scores are low, which shows that the representation of both the agent and the object is inferred correctly (as also shown by the score in our paper). The fact that the score is lower for the agent is expected: the representation of the latter is optimized directly via equivariance w.r.t. actions, while the representation of the object is optimized indirectly by leveraging on the agent. This makes $z_\rm{int}$ easier to infer.
>
>
> **Does this imply that we may not need isometric representations that are translationally shifted from the agent / object states? Perhaps less structured representations are sufficient.**
>
> Our argument is that geometrically structured representations make reinforcement learning (as well as other control and planning problems) more efficient. Even though any (lossless) representation (e.g., a VAE) enables a powerful enough reinforcement learner to eventually solve the task, what we conclude from the reward plot in Figure 4 is that convergence to the solution is significantly faster and more stable with our representation. Indeed, our model stably achieves the highest rewards after around 10k steps, while the VAE starts achieving them after 25k and stabilizes around 50k. This showcases the advantage of geometrically structured representations over generic and less structured ones in terms of efficiency of reinforcement learning.
>
> **Page 12 - "All the conditions imply ... remains equal to z_ext during the execution of actions a1 ... aR" -- why is this the case? After interaction, the object representation could have changed to some z_ext^’ != s_ext^’ + h right?**
>
> After interaction the position $z_\rm{ext}$ can not change because of Condition 3 in Theorem 4.1: by construction $z_\rm{ext}$ does not belong to $\lfloor z_\rm{int} + a_1 + \cdots + a_{i-1}, z_\rm{int} + a_1 + \cdots + a_i \rfloor$ for every $i=1, \cdots, R$ and thus Condition 3 implies that $z_\rm{ext}$ remains constant.

---

### Official Review · Reviewer_M4Jm · 2022-10-30

**Confidence:** 2
**Correctness:** 3
**Technical Novelty And Significance:** 3
**Empirical Novelty And Significance:** 3
**Recommendation:** 8

**Clarity, Quality, Novelty And Reproducibility:**

The paper is well written and I enjoyed the problem setting and the accompanying formulation. The experiments support the claims made in the paper.

**Details Of Ethics Concerns:**

I do not see immediate ethical concerns arising from this work.

**Strength And Weaknesses:**

Strengths:
+ The paper is well written and easy to follow.
+ I like the theoretical framework proposed by the authors and the subsequent learning framework built on top.
+ Proposed experiments confirm the hypothesis put forward by the authors.

Weaknesses:
- Definition 1: \psi is composed of (\psi_{int}, \psi_{ext}).
It is clear that \psi_{int} is equivariant, i.e. z'_{int} = z_{int} + a
But \psi_{ext} doesn't appear to be equivariant.
z'_{ext} != z_{ext} + a
As the state of the object after the contact would depend on the shapes of the object and angle of incidence. How is \psi_{ext}) and equivariant function.
In practice authors move the object at a random location, for the first experiment. Specific to this experiment: Doesn't moving the object to a random location greatly simplify the task, as now we just need to disentangle agent (whose movement follows action) and object (which is moved randomly)? More challenging is the real setting when both agent and object move according to the applied action.

- Based on previous comment, Theorem 4.1 seems to apply to \psi_{int} only and not \psi_{ext}.
This can also be seen in the loss terms. L_{int} only makes \psi_{int} equivariant and L_{ext}+L_{cont} just enforce condition 3 on \psi_{ext}.
There is no term ensuring linearity on \psi_{ext}, which is also not true in the real word.
What makes z_{ext} and z'_{ext} consistent with each other under action a.

- The idea behind eq. 7,8 is to train a proxy encoder \psi_{cont} using contrastive learning and use it's prediction to generate pseudo labels, by partitioning the dataset, to optimise eq. 6. Authors do so because directly optimising eq. 6, collapses the optimisation to one of the terms (L-, L+). Why can't we enforce InfoNCE directly on z_{int} to regularise the space and spread samples in the latent space?

- According to theorem 4.1, if our model has learnt to satisfy the 3 conditions then:
\forall s_i \in D_{test} \psi(w(s_i)) - s_i = h
Why can't we evaluate how much does our model deviate from this constant h.
L_{test} = var(\psi(w(s)) - s). This value should ideally be 0. Why is current metric better?

Clarifications:
- Definition 1: Does the equivariance constrain make the latent representation z_{int} linear?
Does this affect/ limit the properties that the model can capture? Is linear representation space sufficient for complex properties?
In the current formulation the model is reasoning about the translation of the agent and the object, which is linear. Thus enforcing linearity on the latent space makes sense.
What if the model has to learn non-linear properties like rotations? Will it affect the quality of performance?


Minor:
- Caption in Fig. 1: Text uses 's' to describe states of the agent and the object, but the figure caption uses 'z'. 'z' is later used as representation of observation and not the state itself.
- Why consider only translation? Rigid objects can rotate too. Modelling rotations from image observations can be challenging.
- Sec 3: What is n? Is n=3? Translations in 3D Euclidean space?
- Assumption 3.1: (.) is an operator that applies actions 'a' on state 's' to give the new state s'
At the end of paragraph a is applied, using (.) operator, to the observation o. What is the domain of operator (.), state or observation?
- Maybe out of scope for this work but it should be mentioned that state change can happen with a previous contact/interaction. Eg: if you push an object (like a ball) it can continue to move or stop when the contact is removed.

**Summary Of The Paper:**

The paper addresses the task of estimating the states (translation) of the agent and the object that it interacts with using image evidence. Authors assume that the states and properties of the agent and the object are unknown but the actions are known. The key contributions of the work is the formulation that allows decoupling of agent and object features in the latent space.

**Summary Of The Review:**

I'm not an expert in this domain, so I didn't fully follow why is \psi equivariant and not just \psi_{int}. This is important as Theorem 4.1 (the main contribution) requires \psi to be equivariant. See weakness section for details. Apart from this I like the work.

---

> ### Author Response · Authors · 2022-11-10
> **Reply to Reviewer M4Jm (Part 1)**
>
> We thank the reviewer for the appreciation and the extensive review. We would like to comment on the specific points raised.
>
> **Definition 1: \psi is composed of (\psi_{int}, \psi_{ext}). It is clear that \psi_{int} is equivariant, i.e. z'{int} = z{int} + a But \psi_{ext} doesn't appear to be equivariant. z'{ext}!= z{ext} + a As the state of the object after the contact would depend on the shapes of  the object and angle of incidence.**
>
> We agree that Definition 1 might be misleading in its terminology and we have clarified this in the updated version of the manuscript. We call $\varphi$ ‘equivariant’ if $\varphi_\rm{int}$ is translational equivariant w.r.t. actions $a$ i.e., the definition refers to internal states only. This is a choice of terminology and we remark that Definition 1 does not require $\varphi_\rm{ext}$ to be equivariant w.r.t. $a$ since the object is displaced via an unknown transition as correctly pointed out by the reviewer.
>
> **Doesn't moving the object to a random location greatly simplify the task, as now we just need to disentangle agent (whose movement follows action) and object (which is moved randomly)?**
>
> We unfortunately do not see how the (uniformly) stochastic dynamics from the Sprites experiment simplifies the task. What makes the task challenging is that observations are unstructured (images), and thus disentanglement (of agent and object in this case) is challenging – if not unfeasible – via statistics alone (see [1]). Our framework does not rely on any specific statistical aspect of the dynamics and suits arbitrarily complex (and stochastic) dynamics for the object. This is empirically confirmed by the fact that the model performs equally well in terms of the evaluation metric (Equation 10) on both the Sprites and Soccer experiments (see plots in Figure 2 and top-right of Figure 3), which involve radically different dynamics.
>
> [1] Challenging Common Assumptions in the Unsupervised Learning of Disentangled Representations, ICML 2019.
>
> **Based on previous comment, Theorem 4.1 seems to apply to \psi_{int} only and not \psi_{ext}. This can also be seen in the loss terms. L_{int} only makes \psi_{int} equivariant and L_{ext}+L_{cont} just enforce condition 3 on \psi_{ext}. There is no term ensuring linearity on \psi_{ext}, which is also not true in the real world. What makes z_{ext} and z'_{ext} consistent with each other under action a.**
>
> As mentioned by the reviewer, this is related to the first point raised. The statement of Theorem 4.1 is coherent with our choice of terminology (which we expanded upon) in Definition 1: condition 1 (‘$\varphi$ is equivariant) means that $\varphi_\rm{int}$ is translational equivariant w.r.t. actions $a$, and does not require anything from $\varphi_\rm{ext}$.
>
> **Why can't we enforce InfoNCE directly on z_{int} to regularise the space and spread samples in the latent space?**
>
> Unfortunately this question is not clear to us. The component $z_\rm{int}$ is optimized directly via equivariance and thus is already correctly distributed, without the risk of collapse. Thus, we do not see how optimizing the InfoNCE loss for $\varphi_\rm{int}$ would help. We believe that we are missing something here and we would like to ask the reviewer to expand.
>
> **According to theorem 4.1, if our model has learnt to satisfy the 3 conditions then: \forall s_i \in D_{test} \psi(w(s_i)) - s_i = h Why can't we evaluate how much does our model deviate from this constant h. L_{test} = var(\psi(w(s)) - s). This value should ideally be 0. Why is current metric better?**
>
> The score proposed by the reviewer is indeed a valid alternative to the one we use (Equation 10) and has been deployed in various works (see [2]). However, since $s$ is a vector quantity the variance is a matrix quantity (the covariance matrix) and some choices need to be made in order to make the score scalar (e.g., one could use the trace or a matrix norm for this purpose). Moreover, a statistical difference between the two scores is that our is a mean, which is less biased and more stable than a variance.
>
> [2] Tonnar et al., Quantifying and Learning Linear Symmetry-Based Disentanglement, ICML 2022.

---

> ### Author Response · Authors · 2022-11-10
> **Reply to Reviewer M4Jm (Part 2)**
>
> **Does the equivariance constrain make the latent representation z_{int} linear? Does this affect/ limit the properties that the model can capture? Is linear representation space sufficient for complex properties? In the current formulation the model is reasoning about the translation of the agent and the object, which is linear. Thus enforcing linearity on the latent space makes sense. What if the model has to learn non-linear properties like rotations? Will it affect the quality of performance?**
>
> This is a valid point and we discuss it briefly in Section 6 (Limitations and Future Work). While the Euclidean representation indeed suits translations, in general the intrinsic state space corresponds to a \emph{Lie group} $G$, for example the group of rotations $G=\rm{SO}(n)$. Such Lie groups might be non-Euclidean manifolds (as in the case of rotations) and $z_\rm{int}$ would need to live in $G$ instead of the Euclidean space. All of our theoretical results and methods extend to arbitrary Lie groups with assumptions analogous to the ones in Section 3 and 4. Although this represents a relevant and interesting extension, we believe this goes beyond the scope of this work and we leave it for the future.
>
> **Caption in Fig. 1: Text uses 's' to describe states of the agent and the object, but the figure caption uses 'z'. 'z' is later used as representation of observation and not the state itself.**
>
> What we aim to represent in Figure 1 via the turquoise squares is the representation learned by the model $\varphi$ (denoted by arrows). This is why the agent and the object are denoted by the subscripted letter $z$ in the figure, which is coherent with the sections that follow.
>
> **Why consider only translation? Rigid objects can rotate too. Modelling rotations from image observations can be challenging.**
>
> We agree with the reviewer that modeling rotations is important and challenging. Modeling rotations of the agent is related to using Lie groups other than translations. As such, we refer to the answer to the point raised above. As for the object, our framework is actually able to capture shapes and orientations. This is discussed in Section 4.3: to this end, $\varphi_\rm{ext}$ outputs a Gaussian distribution. The (inertia ellipsoid) of the covariance matrix of such a Gaussian represents the shape and  orientation of the object. This is implemented in our model and visualized in the experimental section (see the ellipses in Figure 2 and 3). We agree with the reviewer that this is unclear. Thus, we have rephrased parts of Section 4.3 and emphasized it in Section 3 in the updated version.
>
> **Sec 3: What is n? Is n=3? Translations in 3D Euclidean space?**
>
> Indeed $n$ is the dimension of the ambient Euclidean space ($n=2,3$ in practice).
>
> **Assumption 3.1: (.) is an operator that applies actions 'a' on state 's' to give the new state s' At the end of paragraph a is applied, using (.) operator, to the observation o. What is the domain of operator (.), state or observation?**
>
> We have chosen to abuse the notation and denote by $\cdot$ the transition both in observation space and in state space. This is a common algebraic convention and is done for readability purposes since it avoids overwhelming the notation by tracking the emission map. We agree that this might be unclear and we have remarked this in the updated version.
>
> **Maybe out of scope for this work but it should be mentioned that state change can happen with a previous contact/interaction.**
>
> Indeed our framework assumes that the dynamics are at equilibrium when performing an observation. This is necessary in order to make the state of the object invariant when no interaction occurs. This is how the dataset in the Soccer experiment (Section 5.4) is collected: after performing an action, the agent waits until the ball reaches equilibrium and then records an observation.

---

### Decision · Program_Chairs · 2023-01-20

**Decision:**

Reject

**Justification For Why Not Higher Score:**

The provided theory is only valid for a very constrained setting that involves several strong assumptions. The paper requires significant and fundamental changes to relax these assumptions.

**Justification For Why Not Lower Score:**

N/A

**Metareview: Summary, Strengths And Weaknesses:**

The paper proposes a representation learning framework for the “geometric” state of the agent and objects. The proposed theory shows that an ideal learner disentangles the agent from the object in the representation space. The supervision for representation learning comes from the taken actions (the dynamics is unknown).

Strength:
- The paper provides a theoretical formulation for representation learning (however for a very constrained setting studied in the paper).

- The paper is well-written and it is clear for a wider audience in the ML community.

Weaknesses:
- The paper considers only the “translation” of the agent and the object ignoring all other important aspects of agent-object interaction.

- It assumes the full observability of the environment.

- It hardcodes a policy that forces the agent to interact with objects.

- The experiments have been performed in very simple environments.


**Summary Of Ac-Reviewer Meeting:**

The paper received divergent scores. So the AC had a virtual meeting with all reviewers to discuss the paper. All reviewers appreciated the theoretical framework. However, they had concerns regarding the strong assumptions of the paper (mentioned above) that prevent the applicability of the method to real/more complex scenarios. The AC read the reviews and the rebuttal carefully and agrees that these are strong assumptions. So, the conclusions of the paper might not generalize to even slightly different scenarios that do not conform to these assumptions. The paper is not ready for publication at this point as it needs to relax these strong/unrealistic assumptions. Therefore, the AC recommends rejection.